# SAFE LEARNING THROUGH CONTROLLED EXPANSION OF EXPLORATION SET

## ABSTRACT

Safe reinforcement learning (RL) aims to maximize expected cumulative rewards while satisfying safety constraints, making it well-suited for safety-critical applications. In this paper, we address the setting where the safety of state-action pairs is unknown a priori, with the goal of learning an optimal policy while keeping the learning process as safe as possible. To this end, we propose a novel approach that guarantees almost-sure safety by progressively expanding an exploration set, leveraging previously verified safe state-action pairs and a predictive Gaussian Process (GP) model. We provide theoretical guarantees on asymptotic convergence to the optimal policy and a bound on online regret. Numerical results on benchmark problems with both discrete and continuous state spaces show that our approach achieves superior safety during learning and effectively converges to optimal policies.

## 1 INTRODUCTION

Reinforcement Learning (RL) has demonstrated notable success across diverse applications, including recommendation systems (Afsar et al., 2022). Traditional RL methods, however, primarily focus on expected cumulative reward maximization without adequately considering safety. As a result, unsafe exploratory actions may occur during training, potentially leading to system failures or hazardous outcomes in real-world deployments.

Safe reinforcement learning (Safe RL) methods are broadly categorized into model-based and model-free approaches. Model-based approaches rely on environment models or simulators to predict outcomes and avoid unsafe states. They are often sample-efficient but depend on accurate models of the dynamics and constraints. Constrained Markov Decision Processes (CMDPs) are often used to model problems that minimize cumulative cost under safety constraints. Control-theoretic tools such as Lyapunov functions (Chow et al., 2018) and Model Predictive Control (MPC) (Wabersich & Zeilinger, 2021) have also been employed to enforce stability and safety; for instance, Berkenkamp et al. (2017) used Gaussian processes with Lyapunov analysis to provide stability guarantees. While such approaches can yield strong safety certificates, their reliance on accurate prior models limits applicability in complex uncertain environments.

Model-free approaches avoid explicit models, making them more broadly applicable but typically more data-hungry. A seminal example is Constrained Policy Optimization (CPO, (Achiam et al., 2017)), which adapts policy-gradient methods to CMDPs by constraining updates within a trust region. Although CPO provides worst-case bounds on constraint violations, it is computationally intensive and only enforces constraints in expectation. Other approaches include safety layers that filter or correct unsafe actions at execution time; for example, OptLayer (Pham et al., 2018), which integrates stochastic control principles into neural networks. Similarly, Srinivasan et al. (2020) proposed a safety critic trained from prior tasks to constrain learning in new tasks. These methods intervene online but typically assume safety constraints or cost functions are known a priori.

Beyond the approaches above, a few works study safe exploration when the safety of state–action pairs is not known a priori. Several works focus on gradually expanding a safe set of states or policies. For example, Berkenkamp et al. (2017) considers a deterministic, discrete-time dynamic system and incorporates reachability to fully cover the safely-reachable set while avoiding visits to unsafe pairs with high probability. The algorithm ActSafe that was introduced in As et al. (2024) investigates safe

exploration over feasible policy sets. Such methods rely on iteratively certifying safety and enlarging the exploration space.

Across these lines of work, safety guarantees vary in strength: model-based methods can certify safety given sufficiently structured dynamics (e.g., Marvi & Kiumarsi (2021); Fisac et al. (2019); Liu et al. (2022)), model-free CMDP approaches enforce constraints only in expectation or asymptotically (e.g., Achiam et al. (2017)). Gradually expanding a safe set (e.g., Berkenkamp et al. (2017); As et al. (2024)) can guarantee safety during learning with high probability. For a comprehensive overview, please see Gu et al. (2024).

In contrast to the above, our work aims to address both the following critical challenges:

- Unknown safety of state-action pairs: Many existing approaches often assume that cost functions, safety constraints, or reliable models are given in advance. We consider a harder and more realistic setting where the safety of state-action pairs is unknown a priori, making direct application of CMDP formulations or safety filters infeasible.

- Stronger safety during training: Most prior methods typically tolerate unsafe exploration in the early learning phase as long as the final policy satisfies safety constraints. By contrast, our goal is to keep the entire training process safe, quantified by maintaining unsafe costs below a threshold (i.e., $c(s_t, a_t) \leq \epsilon_{\text{risk}}$) during training.

To address these challenges, we propose a method that starts with an initial safe exploration set and progressively expands it. At each step, we use a Gaussian Process (GP) model to predict the costs of unobserved states and define confidence bounds to certify safe expansion. Policies are trained only within this iteratively updated safe set, ensuring that violations remain tightly controlled. Our work shares the same spirit as As et al. (2024) in terms of gradually enlarging where the agent is allowed to act, but our work differs significantly in safety guarantees and in how safety is defined and certified during learning. More specifically, unlike As et al. (2024) which assumes known costs and defines safety over policies with knowledge of the Lipschitz constant of cost functions, our method defines safety directly over state–action pairs with unknown costs and rewards, leveraging the continuous structure of the environment without requiring Lipschitz constants. Our formulation captures realistic scenarios such as self-driving cars or robotics, where training must be carefully staged to prevent catastrophic failures.

Our main contributions are summarized as follows: (1) We propose a Safe RL algorithm under almost sure safety constraints, which requires every transition to satisfy the safety constraints. (2) We provide theoretical guarantees for the proposed method, including asymptotic convergence to the optimal policy and a finite-episode online regret bound. (3) Through numerical experiments on the Gridworld, CartPole and Safety-Gymnasium benchmarks, we demonstrate that our algorithm achieves superior safety during training and converges effectively to an optimal safe policy.

## 2 PROBLEM FORMULATION

Consider a finite-horizon Constrained Markov Decision Process (CMDP) $\mathcal{M} = (\mathcal{S}, \mathcal{A}, T, \mathcal{P}, \tau, r, c)$ defined over a state space $\mathcal{S}$ and an action space $\mathcal{A}$, with time horizon $T$. For each state $s \in \mathcal{S}$ and action $a \in \mathcal{A}$, the next state $s'$ is drawn from an unknown transition kernel $\mathcal{P}$, i.e., $s' \sim \mathcal{P}(\cdot \mid s, a)$. A non-stationary policy $\pi = \{\pi_n\}_{t=1}^T$ consists of $T$ functions, each mapping states to probability distributions over actions, i.e., $\pi_n : \mathcal{S} \to \Delta(\mathcal{A})$. At each time step $t$, the agent takes an action $a_t$ in state $s_t$, transitions to $s_{t+1}$, receives a reward $r(s_t, a_t)$, and incurs a cost $c(s_t, a_t)$. The initial state $s_1$ is drawn from a known distribution $\tau$. The objective is to maximize the expected cumulative reward while satisfying an almost sure safety constraint:

$$\max_{\pi} \quad \mathbb{E}\left[\sum_{t=1}^T r(s_t, a_t) \mid \pi, s_1 \sim \tau\right] \quad \text{s.t.} \quad c(s_t, a_t) \leq \epsilon_{\text{risk}}, \; \forall t, \tag{1}$$

where $\epsilon_{\text{risk}}$ is a predefined safety threshold. Without loss of generality, we assume that the reward function is bounded as $r(s, a) \in [0, 1]$ to focus on the safety aspects of the problem. In this paper, we consider an online episodic setting, where the agent interacts with the environment over multiple episodes. In each episode, the agent uses data collected in policy deployment from previous episode to both learn the environment and update the policy.

# 3   ALGORITHM

The central idea of our algorithm is to progressively expand the exploration set to ensure each exploration to be safe with a high probability. We predict the safety of unvisited state-action pairs based on previously collected data to define the exploration set. First, we rigorously define the safety for state and state-action pairs.

**Definition 3.1.** *A state $s$ is called safe if there exists at least one safe action $a$ such that $(s, a)$ is safe. Otherwise, the state is considered unsafe.*

We assume access to an initial set $S_0$ containing state-action pairs known to be safe. At each episode, the algorithm proceeds in three stages: (1) Prediction: use a predictive model to estimate the cost and expand the exploration set; (2) Policy Update: update the policy using the data from the previous episode within the newly expanded exploration set; (3) Policy Deployment: execute the updated policy within the new exploration set and collect more data.

## 3.1   PREDICTION

Consider the $n$-th episode. We model the cost function $c(s, a)$ using a Gaussian Process (GP) with mean function $\mu(\cdot)$ and kernel function $k(\cdot, \cdot)$. Assume that the agent observes the cost $c(s, a)$ after experiencing a transition $(s, a, s')$. Denote the cost data collected in episode $n$ as $D_n = (x_i, y_i)_{i=1}^{b_n}$, where $x_i = (s_i, a_i)$, $y_i = c(s_i, a_i)$, and $b_n$ is the batch size. Given the dataset $D_n$, the updated GP posterior mean and variance at a new point $x = (s, a)$ are given by (Seeger, 2004):

$$\mu_{n+1}(x) = \mu_n(x) + \boldsymbol{k}_{x,D_n}^\top \left(\boldsymbol{K}_{D_n,D_n}\right)^{-1} \left(\boldsymbol{y}_{D_n} - \boldsymbol{\mu}_{n,D_n}\right),$$
$$k_{n+1}\left(x, x'\right) = k_n\left(x, x'\right) - \boldsymbol{k}_{\boldsymbol{x},D_n}^\top \left(\boldsymbol{K}_{D_n,D_n}\right)^{-1} \boldsymbol{k}_{\boldsymbol{x}',D_n}, \tag{2}$$

where $k_{x,D_n} = [k_n(x, x_1), \ldots, k_n(x, x_n)]^\top$. The posterior mean represents the predicted value of the cost at $x$, and the posterior variance quantifies the uncertainty in this prediction based on the available data. While we employ GP for its principled uncertainty estimates, other models (e.g., neural networks) may be substituted for prediction provided they can produce a measure of epistemic uncertainty (e.g., predictive variance).

## 3.2   EXPLORATION SET

The objective of this step is to define an exploration set that excludes high-risk state-action pairs, so to ensure learning of the new policy within the exploration set is safe with a high probability.

Using the upper bound of the GP confidence interval, which denotes the pessimistic prediction of the cost, the exploration set at episode $n$ is defined as:

$$S_n = \{(s, a) : \max_a \mu_n(s, a) + \alpha_n k_n(s, a; s, a) \leq \epsilon_{\text{risk}}\}. \tag{3}$$

Here, $\alpha_n$ is a confidence parameter: larger values of $\alpha_n$ correspond to more conservative exploration (i.e., ensuring safety with a higher probability), while smaller values permit more aggressive exploration.

## 3.3   POLICY UPDATE

The objective of this step is to maximize cumulative rewards while ensuring that exploration remains within the exploration set. To this end, we adopt a finite-horizon Q-learning approach. We initialize all Q-values to be zero. For a safe pair $(s, a)$ collected in the previous episode, we set the terminal-stage Q-value as $Q_T^n(s, a) = r(s, a)$ in episode $n$, and perform the following backward update:

$$Q_t^n(s, a) \leftarrow (1 - \beta_n) Q_t^{n-1}(s, a) + \beta_n \left[ r_t(s, a) + \max_{a' \in \mathcal{A}} Q_{t+1}^n\left(s', a'\right) \right], t = T - 1, \ldots, 1 : \tag{4}$$

where $\beta_n$ is the learning rate at episode $n$. For any unsafe transition $(s, a)$ observed in the previous episode, we set $Q_t^n(s, a) = -1$ for all $t$, to discourage unsafe exploration, and update $S_n$ to exclude these pairs. To encourage sufficient exploration within the exploration set, we augment the Q-values

with an Upper Confidence Bound (UCB) bonus term. Specifically, for state-action pairs $(s, a) \in S_n$, we define:

$$\tilde{Q}_t^n(s, a) = Q_t^n(s, a) + c \cdot \sqrt{\frac{\log(L_n)}{N_n(s, a)}}, \tag{5}$$

where $L_n$ is the total number of transitions observed so far, $N_n(s, a)$ is the number of times $(s, a)$ has been visited, and $c$ is a positive constant. For pairs outside the exploration set, we set $\tilde{Q}_t^n(s, a) = Q_t^n(s, a)$. A greedy policy is then derived from the Q-values with exploration bonuses, selecting the action $\arg\max_{a \in \mathcal{A}} \tilde{Q}_t^n(s, a)$ for state $s$. To ensure sufficient exploration on the boundary of the exploration set, we define the final policy $\pi_n$ as an $\epsilon_p$-greedy policy: with probability $1 - \epsilon_p$, the greedy action is selected, and with probability $\epsilon_p$, a random safe or unexplored action is chosen.

### 3.4 Policy Deployment

**Definition 3.2.** *If an unsafe state-action pair $(s, a)$ has been visited, then we say this pair is verified to be unsafe. If all actions for a state have been verified to be unsafe, we say this state is verified to be unsafe.*

The policy $\pi_n$ is deployed within the exploration set $S_n$ to collect new dataset $D_n$. Define $B_n$ to be the set of states that has been visited and not been verified to be unsafe before episode $n$. Each trajectory in policy deployment starts randomly at any state in $B_n$ and terminates if the current state-action pair is outside $S_n$, or the length of trajectory is larger than $T$, or safety constraint is violated more than once. After data collection, the GP model is updated according to equation 2 using the newly acquired dataset $D_n$. For the starting state, a straightforward approach is to initialize each trajectory by following uniform distribution, which also helps theoretical analysis later. In practice, we can use a non-uniform initialization scheme that prioritizes less-explored states. Specifically, the probability of starting at state $s$ can be set proportional to $e^{-\kappa N_n(s)}$, where $N_n(s)$ denotes the number of transitions to state $s$ observed so far and $\kappa$ is a constant. The full algorithm is presented in Algorithm1 below.

---

**Algorithm 1** Learning by Safe Expansion of Exploration Set(LearnSEES)

---

1: **Input:** An initial safe exploration set $S_0$ and an initial safe policy $\pi_0$, the safety threshold $\epsilon_{risk}$ .
2: Deploy the policy $\pi_0$ in the exploration set $S_0$ to obtain the initial dataset $D_0$. If the agent move to the outside of $S_0$ or the length of trajectory is larger than $T$, stop.
3: **for** n=1 to N **do**
4:     Update GP to $(\mu_n, k_n)$ by equation 2 based on $D_{n-1}$.
5:     **Expanding Exploration Set:** Define $S_n$ according to equation 3.
6:     **Policy Training:** For a transition $(s, a, s')$ in $D_{n-1}$,
7:     **if** the current transition $(s, a, s')$ is safe **then**
8:         For $t = T - 1, T - 1, \ldots, 1$, do Q-learning update according to equation 4
9:     **else**
10:         Set $Q_t(s, a) = -1$ for all $t$.
11:     **end if**
12:     For pairs $(s, a)$ in $S_n$, excluding verified unsafe pairs, define $\tilde{Q}_t^n(s, a)$ with the UCB bonus according to equation 5. Otherwise, set $\tilde{Q}_t^n(s, a) = Q_t^n(s, a)$.
13:     Calculate the $\epsilon_p$-greedy policy $\pi_n$: use the greedy policy based on $\tilde{Q}_t^n$ with probability $1 - \epsilon_p$, or uniformly randomly pick a safe or unexplored action with total probability $\epsilon_p$ .
14:     **Data Collection**: Deploy policy $\pi_n$ in true environment to collect new dataset $D_n$ with the starting and terminating rule described in Section 3.4 .
15: **end for**
16: **Output**: policy $\pi_N$.

---

## 4 Theoretical Analysis

In this section, we analyze the convergence properties of Algorithm 1. We begin by outlining key assumptions, then establish asymptotic convergence guarantees, and finally provide a finite-episode upper bound on the online regret.

## 4.1 ASYMPTOTICAL CONVERGENCE

Algorithm 1 involves two intertwined convergence processes. First, the **exploration set** converges: as the algorithm expands the exploration set and identifies unsafe pairs over time, it is expected to eventually encompass all safe state-action pairs that are reachable from the initial state. Second, the **policy** converges: since policy updates and deployments are restricted to the current exploration set, the learned policy is anticipated to converge to the optimal policy within the sub-MDP induced by the safe pairs. A key analytical challenge in our algorithm lies in characterizing the number of episodes required to visit and accurately learn the cost associated with all relevant state-action pairs. We now formalize these intuitions by making following assumptions.

**Assumption 4.1.** *The state space $\mathcal{S}$ and action space $\mathcal{A}$ are finite.*

It should be noticed that Algorithm 1 also works for continuous-state problems in practice, as we illustrated in the numerical section 5.2 and 5.3. To make the problem feasible, there should exist at least one trajectory consisting of safe pairs.

**Assumption 4.2.** *For any two safe states that are reachable from a common initial state, there exists a trajectory composed entirely of safe pairs connecting them. Furthermore, for every safe transition $(s, a, s')$, there exists at least one safe action available at $s'$.*

Then we show asymptotic convergence of Algorithm 1.

**Theorem 1** (Asymptotic Convergence). *Suppose Assumptions 4.1 and 4.2 hold. Then, the policy sequence generated by Algorithm 1 converges almost surely to the optimal policy as the number of episodes $n \to \infty$.*

The proof of Theorem Theorem 1 is provided in Appendix B.1. Let $\bar{M} = (\bar{\mathcal{S}}, \{\bar{\mathcal{A}}_s\}_{s \in \bar{\mathcal{S}}}, \bar{P}, \bar{r})$ denote the safe sub-MDP, where $\bar{\mathcal{S}}$ is the set of all safe states, $\bar{\mathcal{A}}_s$ is the set of all safe actions for state $s \in \bar{\mathcal{S}}$, and $\bar{P}$ and $\bar{r}$ are the restrictions of the original transition kernel and reward function to $\bar{\mathcal{S}}$ and each $\bar{\mathcal{A}}_s$. Theorem 1 establishes that our algorithm converges almost surely to the optimal policy over the safe sub-MDP $\bar{M}$. It is important to note that violations of safety constraints may still occur during policy training as part of the necessary tradeoff to enable exploration; however, the violations are kept under the specified threshold $\epsilon_{risk}$ in each episode.

## 4.2 ONLINE REGRET

For an online reinforcement learning problem, it is crucial to quantify not only asymptotic convergence but also the finite-time performance of the algorithm. In particular, we study the cumulative *regret* incurred over $N$ episodes, defined as:

$$\text{Regret}(N) = \mathbb{E}\left[\sum_{n=1}^{N} \left(V_1^*(s_{n,1}) - V_1^{\pi_n}(s_{n,1})\right)\right],$$

where $V_1^*$ is the value function at the first time stage under the optimal policy , $V_1^{\pi_n}$ is the value function under the policy $\pi_n$ used in episode $n$, and $s_{n,1}$ is the initial state in episode $n$.

The regret analysis can be decomposed into two parts:(1) The regret incurred before all safe state-action pairs are identified, i.e., during the convergence of the exploration set; (2) The regret incurred afterwards, which corresponds to the online performance of Q-learning over the induced safe sub-MDP. We now present the main result on the regret bound.

**Theorem 2** (Online Regret Bound). *Suppose Assumptions 4.1 and 4.2 hold. Then the expected cumulative regret after $N$ episodes is bounded by*

$$\text{Regret}(N) \leq T \cdot \frac{|S|^2|A|^2}{\epsilon_p \delta} + \mathcal{O}\left(\frac{T^6 SA}{\Delta_{\min}} \log(SANT)\right),$$

*where $\Delta_t(x, a) := V_t^*(x) - Q_t^*(x, a)$, $\Delta_{\min} = \min_{t,x,a} \{\Delta_t(x, a) : \Delta_t(x, a) \neq 0\}$ is the minimum non-zero suboptimality gap over all safe state-action pairs, $T$ is the episodic horizon length, $\delta := \min\{p(s'|s, a) : p(s'|s, a) > 0\}$ is the lower bound of all positive transition probabilities.*

The proof of Theorem 2 is provided in Appendix B.2. Theorem 2 establishes that the online regret consists of two components: a constant term and a logarithmic term in the number of episodes $N$.

The constant term arises from the cost of learning the safe structure of the entire state-action space, which must occur before safe Q-learning can be reliably applied. The logarithmic term reflects the logarithmic regret incurred by Q-learning over the safe sub-MDP, once the exploration set has converged. The logarithmic regret for regular Q-learning is proved by Yang et al. (2021). To the best of our knowledge, this $\mathcal{O}(\log N)$ dependence on the number of episodes represents the best-known regret bound for safe RL with almost sure constraint. For regret analysis in CMDPs, (Efroni et al., 2020) derives $\tilde{O}\left(N^{1/2}\right)$ regret and violation bounds for episodic CMDPs via an optimistic UCB-style algorithm. While still allowing sublinear cumulative constraint violations, (Ding et al., 2022) proves the regret bound to $\tilde{O}(N^{1/2})$ for Natural Policy Gradient Primal-Dual Method. The first primal-dual algorithm achieving a sublinear regret guarantee without allowing error cancellations is proposed by (Müller et al., 2024).

## 5 NUMERICAL EXPERIMENTS

We evaluate our proposed algorithm LearnSEES on three benchmark problems: Gridworld, CartPole and Safety-Gymanasium. In the Gridworld problem, we compare our method against a linear search baseline; in CartPole, we compare our method against the Deep Q-Network (DQN) algorithm and ActSafe; and in Safety-Gymnasium, we compare our method against Constrained Policy Optimization (CPO). In some of these problems, we extend our algorithm to handle continuous states and continuous actions. We also numerically validate our theoretical results and show the convergence of our algorithm to an optimal safe policy.

### 5.1 DISCRETE GRIDWORLD

We first validate our algorithm using a modified Gridworld benchmark inspired by Leike et al. (2017). Specifically, we consider a 12×12 grid, as illustrated in Figure 1. The agent begins at the grid marked 'S' and aims to reach the goal 'G' as fast as possible while avoiding regions associated with high risks. States correspond to grid coordinates, numbered from $0$ to $143$. Actions consist of discrete directional moves: $(\pm 1/3, \pm 2/3)$ or $(\pm 2/3, \pm 1/3)$, where each action probabilistically determines horizontal and vertical movements and the sign determines the direction. The incurred cost depends solely on the current grid state, i.e., $c(s, a) = c(s)$, which is shown by the numbers in Figure 1. Therefore, blue grids are unsafe states, and white grids are safe states. The agent receives reward $T$ when it reaches the goal, $T/2$ per step when it stays at the goal after reaching there, and zero otherwise. Three distinct safety thresholds are considered: $\epsilon_{\text{risk}} \in \{0.25, 0.75, 1.25\}$. Initially, the exploration set $S_0$ comprises states $\{0, 1, 2, 3, 4, 5\}$, with Gaussian Process (GP) initialized using cost observations from the first three rows and six columns.

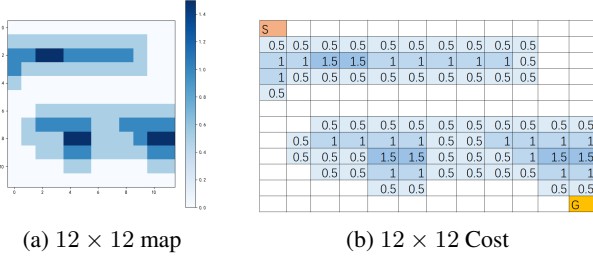

(a) $12 \times 12$ map        (b) $12 \times 12$ Cost

Figure 1: Setting for Gridworld problem.

The results for $\epsilon_{\text{risk}} = 0.75$ is demonstrated in Figure 2, and the results for $\epsilon_{\text{risk}} = 0.25$ and $1.25$ are shown in Appendix B.3.1. Figure 2a shows the number of visiting each state for the whole state spaces, which shows that the region near the optimal route has been visited more often than other states. Figure 2b shows the route generated by the final policy, which shows that our algorithm can learn a nearly optimal policy as well as learn the cost setting adjacent to the optimal route. We also conduct linear search method as a baseline, which traverses all state-action pairs in each iteration and exclude pairs having unsafe history. As shown in Figure 2c, linear search also finds a nearly optimal route at the end. However, compared to linear search, our algorithm achieves a significantly lower violation rate (0.43% vs. 2.26%), which is the percentage of transitions where

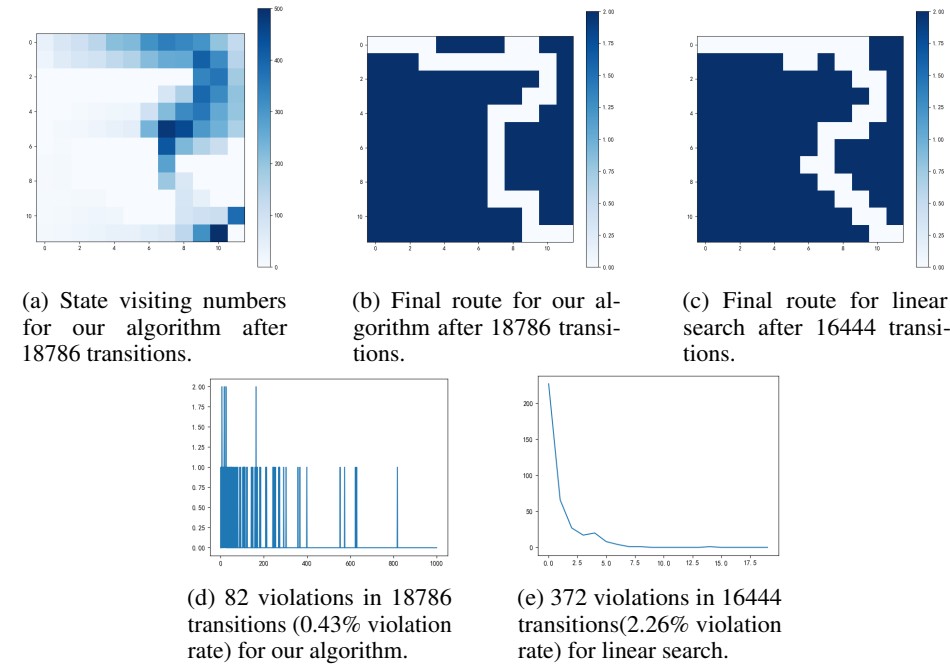

(a) State visiting numbers for our algorithm after 18786 transitions.

(b) Final route for our algorithm after 18786 transitions.

(c) Final route for linear search after 16444 transitions.

(d) 82 violations in 18786 transitions (0.43% violation rate) for our algorithm.

(e) 372 violations in 16444 transitions(2.26% violation rate) for linear search.

Figure 2: State visiting and constraint violation for $\epsilon_{risk} = 0.75$.

the per-step cost exceeds the safety threshold, preventing large risks—particularly during the early training phase. While linear search incurs many violations in the initial exploration period and exhibits a higher overall violation rate, our method distributes violations more evenly throughout training. This comparison is illustrated in Figure 2d and Figure 2e. Figure 2d shows the violation rate for one typical replication of our algorithm. Running 10 replications generates the violation rates with mean 0.43 and standard error $6.5 \times 10^{-4}$, showing the stability of our algorithm across different replications. This comparison demonstrates that our algorithm is safer during training process, which is consistent with the intuition and theoretical analysis.

## 5.2 CONTINUOUS GRIDWORLD

Assumption 4.1, which is about finite state and action spaces, is made for theoretical convergence. In practice, our algorithm can be applied to problem with continuous state and action spaces. Next, we extend our evaluation to a continuous analog of the discrete Gridworld. The environment consists of a 10×10 square with the agent aiming to reach a goal region in the upper-right corner while avoiding two dangerous circular areas, which is showed in Figure 3a. The action space includes movements in four cardinal directions, with the actual movement distance drawn uniformly from $U(0.5, 1)$. For a transition $(s, a, s')$, the cost function penalizes proximity to the dangerous circles, defined as $\max(4 - 2d_1, 4 - 2d_2, 0)$, where $d_1$ and $d_2$ denote Euclidean distances from the agent's next position $s'$ to the centers of two hazardous circles located at $(3, 3)$ and $(7, 7)$ with radius 2. Reward is defined as $\max\{10(1 - dist(s, (10, 10)), 0\}$, which is positive exclusively in the upper-right corner and diminishes with distance from the goal. Algorithm 1 was adapted to handle continuous state space: discretizing state space and using the grid centers as representative points for GP predictions, training policy and Q-values using neural networks, and employing a modified reward structure incorporating a penalty term proportional to the incurred cost. Details are included in Appendix B.3.2. Figure 3b visualizes the data distribution in a typical replication of the algorithm run, which achieves a low violation rate of 2.3%. The violation rates over 10 replications have mean 2.0% and standard error $7.0 \times 10^{-4}$, showing the stability of our algorithm across different replications. Figure 3c shows that our algorithm can find a nearly optimal route at the end.

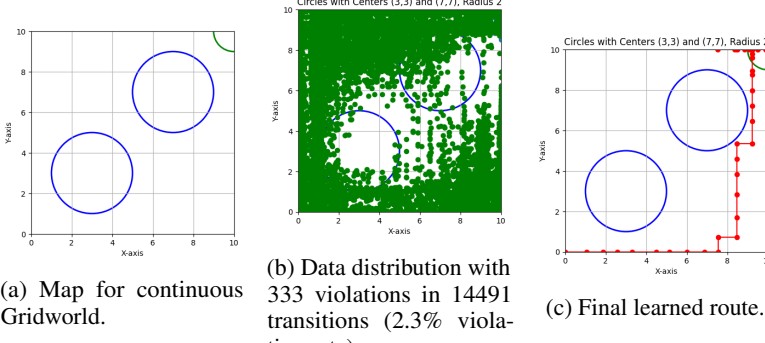

(a) Map for continuous Gridworld.

(b) Data distribution with 333 violations in 14491 transitions (2.3% violation rate).

(c) Final learned route.

Figure 3: Environment Setting, data distribution and the final learned route for continuous Gridworld.

## 5.3 CARTPOLE

We test our algorithm on the classic CartPole task, comparing against Deep Q-Network (DQN) (Mnih et al., 2015) and ActSafe (As et al., 2024). CartPole environment considers a pole attached by an un-actuated joint to a cart moving along a frictionless track, where the goal is to apply horizontal forces to the cart to keep the pole balanced upright as long as possible. DQN approximates the Q-value function using a neural network, and we adapt DQN to use the penalized reward $r'(s,a) = r(s,a) - 50 \cdot c(s,a,s')$ in consideration of safety. ActSafe iteratively learns a probabilistic model (e.g., a Gaussian process) of the environment's transition dynamics from previously collected data, update the policy by optimizing the worst-case performance over a set of policies that defined to be safe with high probability, and deploy the policy to collect new data. While ActSafe needs to assume known reward and cost to make policy decision and considers the total cost constraint, our setting considers unknown reward and cost and consider the per-step cost constraint. So, we have to use different settings for two algorithms, and our setting of unknown cost/reward is much more challenging than ActSafe. Besides, ActSafe considers an accumulated cost constraint, so we have to adjust ActSafe's setup in implementation to an average cost constraint for fair comparison.

We set $T = 200$ and the safety threshold $\epsilon_{\text{risk}} = 0.3$. Denote $p$ and $w$ to be the position and pole angle of the pole. Then the cost is define to be $\max\{\text{dist}(p, [-1.9, 1.9]), 10 \cdot \text{dist}(w, [-0.15, 0.15])\}$, which is maximum distance of the pole position and pole angle (with the scaling parameter 10) away from a safe interval. We show the training violation rate and the average performance of final policy over 100 testing trajectories in Table 1. More setting details are provided in Appendix B.3.3. Our method consistently demonstrates fewer violations during training relative to DQN and ensures absolute safety for the final policy. Compared to ActSafe, our method achieves a lower ratio of violations throughout training. For the performance of the final policy, ActSafe and our algorithm can both converge to an absolutely safe optimal policy. Over 10 replications of our algorithm, the training violation rates have mean 0.01 and standard error $4 \times 10^{-4}$, the average total rewards have mean 199.8 and standard error 0.099, and the testing violation rates have mean $1.2 \times 10^{-4}$ and standard error $1.5 \times 10^{-7}$. In 9 out of 10 replications, our algorithm achieved the maximum total reward and zero testing violations, showing superior optimality and safety of our algorithm.

Table 1: CartPole Results: For training, we show the violation rate. For the testing of final policy, we show the total reward and violation rate for the testing performance over 100 trajectories.

|  | Training Violation Rate | Total Reward | Testing Violation Rate |
|---|---|---|---|
| Our Algorithm | 0.008 | 200 | 0 |
| DQN | 0.018 | 200 | 0.0017 |
| ActSafe | 0.0258 | 200 | 0 |

## 5.4 SAFETY-GYMNASIUM

We evaluate on Safety-Gymnasium (Ji et al., 2023), focusing on SafetyPointGoal1-v0, a continuous-control task with a 60-dimensional continuous state and a 2-dimensional continuous action. The

agent is a velocity-controlled point robot in a 2-D workspace that must reach a moving goal while avoiding circular keep-out zones ("hazards"). Details about environment setting are provided in Appendix B.3.4. Because both state and action are continuous, we replace the GP in Algorithm 1 with a probabilistic cost model implemented by neural networks (NNs) that predicts a mean $\mu_n(s, a)$ and variance $\sigma_n^2(s, a)$ for $c(s, a)$. For policy training, we employ a parametric Q-learning controller to select actions, updating it using the classical Q-learning rule.

Just like in Algorithm 1, action is selected according to the cost model: we act only with actions whose upper confidence bound (UCB) on predicted cost is nonpositive, i.e. $\mu_n(s, a) + \alpha_n \sigma_n(s, a) \le \varepsilon_{\text{risk}}$, otherwise we choose the action with the minimal UCB cost. This strategy preserves safety early in training while expanding coverage as uncertainty diminishes. Full implementation details are listed in Appendix B.3.4.

We compare against Constrained Policy Optimization (CPO) (Achiam et al., 2017), a classical Safe RL algorithm. Figure 4 reports learning curves for return and cost, along with cost histograms for both methods on a typical run. Compared with CPO, our method yields substantially lower training costs and higher final rewards, as well as much less violation during the training process.

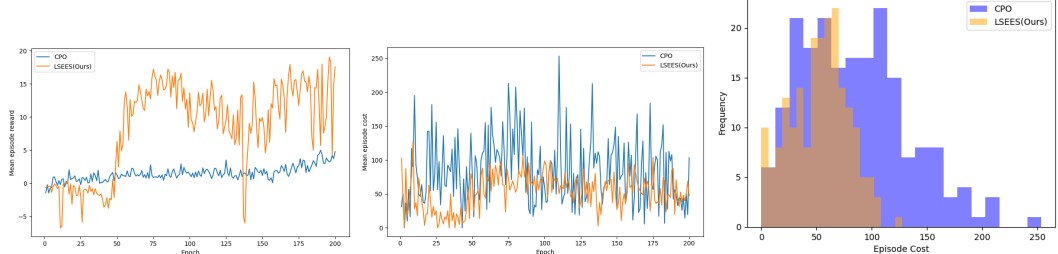

Figure 4: **Safety-Gymnasium PointGoal1.** Left: average discounted returns per episode vs. the number of episodes. Middle: average discounted costs per episode vs. the number of episodes. Right: Per-episode cost hisograms.

## 6 CONCLUSION

In this paper, we proposed a novel model-free Safe RL algorithm, designed to reduce unsafe exploration during training and to converge to an optimal safe policy. The algorithm progressively expands an exploration set of state-action pairs, leveraging Gaussian Process models for cost estimation based on visited state-action pairs. In each episode, it guides exploration cautiously while enabling policy improvement within the verified safe region. We provided theoretical guarantees, establishing both asymptotic convergence to the optimal safe policy and a finite-episode upper bound on online regret. Empirical results in discrete and continuous environments, including Gridworld, CartPole, and Safety-Gymanasium, demonstrate the effectiveness of our approach in maintaining safety during training and achieving near-optimal performance in the final policy.

## REPRODUCIBILITY STATEMENT

Most implementation and experimental details required to replicate our results are provided in the Appendix B.3. The anonymized supplementary materials include our complete codes, which contain the remaining experimental details.

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

## A    THE USE OF LARGE LANGUAGE MODELS (LLMs)

In this work, the large language model (LLM) was used for text grammar refinement, searching related literature and code debugging.

## B    TECHNICAL APPENDICES AND SUPPLEMENTARY MATERIAL

### B.1    PROOF OF THEOREM 1

*Proof.* Suppose that not all transitions have been visited before episode $n$. Recall that $B_n$ defined in Section 3.4 is the set of states that has been visited and not been verified to be unsafe before episode $n$, and a new trajectory at episode $n$ begins randomly and uniformly at an initial state $\bar{s} \in B_n$. There are two cases:(1) All actions for $\bar{s}$ have been visited and there exists at least one action $\bar{a}$ that $(\bar{s}, \bar{a})$ is safe, and all safe next states reachable from $\bar{s}$ by one safe transition have been visited ; (2) All actions for $\bar{s}$ have been visited and there exists at least one action $\bar{a}$ that $(\bar{s}, \bar{a})$ is safe, and at least one safe state reachable from $\bar{s}$ by one safe transition has not been visited ; (3) At least one action for $\bar{s}$ has not been visited. By Assumption 4.2, the probability of case (2) or case (3) is zero if and only if we have explored all safe state-action pairs reachable from the initial state. Thus, the probability of case (2) or case (3) is nonzero, and at least $\frac{1}{|B_n|}$.

Denote $T_i$ as the number of episodes between visiting the $i$-th and $(i-1)$-the new state-action pairs, and let $M = \sum_{i=1}^{\infty} T_i$. Here, $M$ is actually a finite summation of $T_i$'s because the number of state-action pairs is finite. Define $\delta := \min\{p(s'|s, a) : p(s'|s, a) > 0\}$ to be the lower bound of all positive transition probabilities. In each episode, the probability of visiting a new state-action pair is at least $\frac{\delta \epsilon_p}{|B_n||\mathcal{A}|}$ by $\epsilon_p$-greedy policy. Assuming the probability of visiting a new state-action pair is exactly $\frac{\delta \epsilon_p}{|B_n||\mathcal{A}|}$ in each episode, then the number of episodes needed to visit a new transition follows a geometric distribution with success probability $p = \frac{\delta \epsilon_p}{|B_n||\mathcal{A}|}$. Thus, $\mathbb{E}[T_i]$ is smaller than the expectation of such a geometric random variable, i.e., $\mathbb{E}[T_i] \leq \frac{|B_n||\mathcal{A}|}{\epsilon_p \delta} \leq \frac{|\mathcal{S}||\mathcal{A}|}{\epsilon_p \delta}$. Summing over all pairs(at most $|S||A|$), we obtain $\mathbb{E}[M] \leq \frac{|S|^2|A|^2}{\epsilon_p \delta}$. Therefore, $S_n$ will contain all safe state-action pairs after a finite number of episodes with probability 1. Let $\bar{M} = (\bar{\mathcal{S}}, \{\bar{\mathcal{A}}_s\}_{s \in \bar{\mathcal{S}}}, \bar{P}, \bar{r})$ denote the safe sub-MDP, where $\bar{\mathcal{S}}$ is the set of all safe states, $\bar{\mathcal{A}}_s$ is the set of all safe actions for state $s \in \bar{\mathcal{S}}$, and $\bar{P}$ and $\bar{r}$ are the restrictions of the original transition kernel and reward function to $\bar{\mathcal{S}}$ and each $\bar{\mathcal{A}}_s$. By Definition 3.1 and Assumption 4.2, for any $s \in \bar{\mathcal{S}}$ and $a \in \bar{\mathcal{A}}_s$, $(s, a)$ only transit to states in $\bar{\mathcal{S}}$. Thus, $\bar{P}$ and $\bar{r}$ are well-defined.

After $S_n$ contains all safe pairs, the policy will generate trajectories that remain entirely within $\bar{M}$. Moreover, every safe pair is visited infinitely often due to the $\epsilon_p$-greedy policy. Thus, $Q_n|_{\bar{M}}$ converges almost surely to $\bar{Q}^*$, the optimal Q-function for $\bar{M}$. Then, by Theorem 3 of VP & Bhatnagar (2021), we have: (1) $\sup_n \|Q_n|_{\bar{M}}\| < \infty$ almost surely; (2) $Q_n|_{\bar{M}} \to \bar{Q}^*$ as $n \to \infty$ almost surely. $\qquad \square$

### B.2    PROOF OF THEOREM 2

Define the constants    $\alpha_t = \frac{T+1}{T+t}$,    $\alpha_t^0 = \prod_{j=1}^{t}(1 - \alpha_j)$ and $\alpha_t^i = \alpha_i \prod_{j=i+1}^{t}(1 - \alpha_j)\,(i > 0)$. Let $\beta_0 = 0$ and $\beta_t = 4c\sqrt{\frac{T^3\iota}{t}}$ for $t \geq 1$ and the event

$$\mathcal{E}_{\text{conc}} := \{\forall(s, a, t, n) : 0 \leq (Q_t^n - Q_t^*)(s, a) \leq$$

$$\alpha_{n_h^n}^0 T + \sum_{i=1}^{n_t^n} \alpha_{n_t^n}^i \left(V_{t+1}^{\tau_t(s,a,i)} - V^*\right)\left(s_{t+1}^{\tau_t(s,a,i)}\right) + \beta_{n_t^n}\Bigg\}.$$

We first show two lemmas that will be used.

**Lemma B.1.** *Lemma 4.1 (Concentration)Yang et al. (2021). Event $\mathcal{E}_{conc}$ occurs w.p. at least $1 - 1/NT$.*

**Lemma B.2.** *Lemma 4.2 (Bounded Number of Steps in Each Interval)Yang et al. (2021). Under $\mathcal{E}_{conc}$, we have for every $n \in [N]$,*

$$C^{(n)} := \left| \left\{ (k,h) : \begin{array}{c} \left(Q_t^k - Q_t^*\right)\left(s_t^k, a_t^k\right) \in \\ \left[2^{n-1}\Delta_{\min}, 2^n\Delta_{\min}\right) \end{array} \right\} \right|$$

$$\leq \mathcal{O}\left(\frac{T^6 SA\iota}{4^n \Delta_{\min}^2}\right), \quad \text{where } \iota = \log\left(SAN^2 T^2\right)$$

**Proof of Theorem 2**

*Proof.* Define $N_s$ to be the time that the agent has visited each state-action pairs at least once. We know from the proof of Theorem 1 that $\mathbb{E}[N_s] \leq \frac{|S|^2|A|^2}{\epsilon_p \delta}$, thus $N_s$ is finite with w.p.1. Notice that $N_s$ is a stopping time for MDP $\mathcal{M}$.

$$\text{Regert}(N) = \mathbb{E}\left[\sum_{n=1}^{N_s}\left(V_1^*\left(s_{n,1}\right) - V_1^{\pi_n}\left(s_{n,1}\right)\right)\right] + \mathbb{E}\left[\sum_{n=N_s+1}^{N}\left(V_1^*\left(s_{n,1}\right) - V_1^{\pi_n}\left(s_{n,1}\right)\right)\right]$$

$$= I + II$$

For *I*, as each reward is assumed to be in $[0,1]$, the value function gap $V_1^*\left(s_{n,1}\right) - V_1^{\pi_n}\left(s_{n,1}\right)$ is bounded by the time horizon $T$. Then we can get a bound for $I$ by

$$I = \mathbb{E}\left[\sum_{n=1}^{N_s}\left(V_1^*\left(s_{n,1}\right) - V_1^{\pi_n}\left(s_{n,1}\right)\right)\right] \leq T\mathbb{E}[N_s] \leq \frac{T|S|^2|A|^2}{\epsilon_p \delta}$$

For *II*, we now only need to consider consider the case $n > N_s$. Notice that if we have visited all state-action pairs at least once, we will only choose safe actions. Equivalently, we are doing the Q-learning with UCB for a safe sub-MDP $\overline{M}$ that only contains all safe actions from the original unconstrained MDP $M$, which appears in the proof of Theorem 1.

For the upper bound of $II$, we follows the proof of Theorem 3.1 Yang et al. (2021). First, we can construct a recursive equation.

$$\left(V_1^* - V_1^{\pi^n}\right)\left(s_1^n\right)$$

$$= V_1^*\left(s_1^n\right) - Q_1^*\left(s_1^n, a_1^n\right) + \left(Q_1^* - Q_1^{\pi^n}\right)\left(s_1^n, a_1^n\right)$$

$$= \Delta_1\left(s_1^n, a_1^n\right) + \mathbb{E}_{s' \sim P_1\left(\cdot | s_1^n, a_1^n\right)}\left[\left(V_2^* - V_2^{\pi^n}\right)\left(s'\right)\right]$$

$$= \cdots = \mathbb{E}\left[\sum_{t=1}^{T}\Delta_t\left(s_t^n, a_t^n\right) \mid a_t^n = \pi_n\left(s_t^n\right)\right].$$

Notice that $V_t^*\left(s_t^n\right) = Q_t^*\left(s_t^n, a^*\right) \leq Q_t^n\left(s_t^n, a^*\right) \leq Q_t^n\left(s_t^n, a_t^n\right)$. Then we have

$$\Delta_t\left(s_t^n, a_t^n\right) = \text{clip}\left[V_t^*\left(s_t^n\right) - Q_t^*\left(s_t^n, a_t^n\right) \mid \Delta_{\min}\right]$$
$$\leq \text{clip}\left[\left(Q_t^n - Q_t^*\right)\left(s_t^n, a_t^n\right) \mid \Delta_{\min}\right],$$

where clip is defined as $\text{clip}[s \mid \delta] := s \cdot \mathbb{I}[s \geq \delta]$ and $\Delta_{\min}$ is the minimum non-zero gap: $\Delta_{\min} := \min_{s,a}\{\Delta(s,a) : \Delta(s,a) \neq 0\}$.

Finally, by using Lemma B.1 and Lemma B.2, we have

$$
\begin{aligned}
II =& \mathbb{E}\left[\sum_{n=N_s+1}^{N}\sum_{t=1}^{T}\Delta_t\left(s_t^n, a_t^n\right)\right] \\
=& \sum_{\text{traj}}\mathbb{P}(\text{ traj })\cdot\sum_{n,t}\Delta_t\left(s_t^n, a_t^n \mid \text{ traj }\right) \\
\leq& \sum_{\text{traj }\in\mathcal{E}_{\text{conc}}}\mathbb{P}(\text{ traj })\cdot\sum_{n,t}\text{clip}\left[\left(Q_t^n - Q_t^*\right)\left(s_t^n, a_t^n \mid \text{ traj }\right)\mid\Delta_{\min}\right] \\
& + \sum_{\text{traj }\in\overline{\mathcal{E}_{\text{conc}}}}\mathbb{P}(\text{ traj })\cdot NT\cdot T \\
\leq& \mathbb{P}\left(\mathcal{E}_{\text{conc}}\right)\sum_{n=1}^{N}2^n\Delta_{\min}C^{(n)} + \mathbb{P}\left(\overline{\mathcal{E}_{\text{conc}}}\right)\cdot NT\cdot T \\
\leq& \sum_{n=1}^{N}\mathcal{O}\left(\frac{T^6 SAL}{2^n\Delta_{\min}}\right) + T \\
\leq& \mathcal{O}\left(\frac{T^6 SA}{\Delta_{\min}}\log(SANT)\right).
\end{aligned}
$$

Combining the bounds for $I$ and $II$, we complete the proof. □

### B.3    DETAILS FOR NUMERICAL EXPERIMENTS

#### B.3.1    DISCRETE GRIDWORLD

The detailed parameters are set as below: time horizon $T = 80$, total episode number $N = 1000$, the UCB coefficient for Q-value function is set to 20. Besides the results for $\epsilon_{\text{risk}} = 0.75$ in Section 5.1, we also show the results for $\epsilon_{\text{risk}} = 0.25$ and $1.25$ here. For $\epsilon_{\text{risk}} = 0.75$ or $1.25$, we use the uniform starting principle. Fo r$\epsilon_{\text{risk}} = 0.25$, as the region of safe states is small compared with unsafe states, we set the starting probability at $s$ to be proportional to $\exp(-2N(s))$ to accelerating exploration. Figure 5 and 6 shows that our algorithm can converge to a nearly optimal policy and experience small vioaltion rates in different settings.

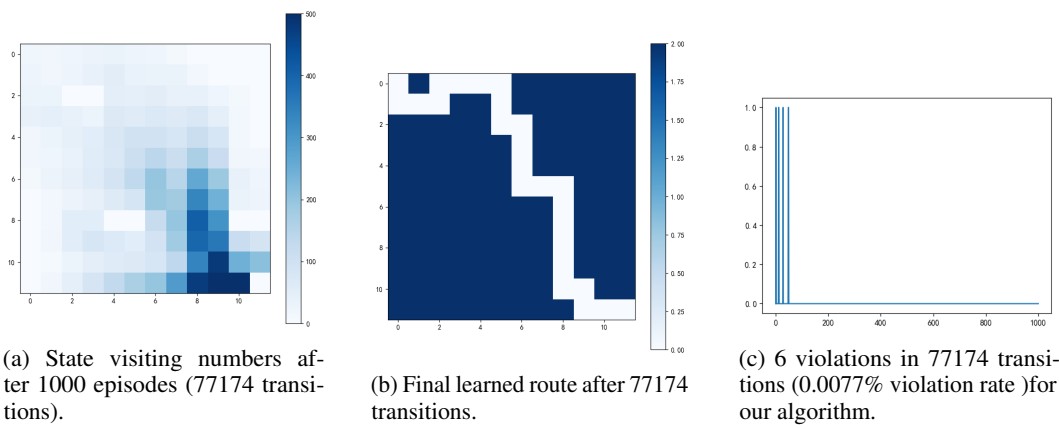

(a) State visiting numbers after 1000 episodes (77174 transitions).

(b) Final learned route after 77174 transitions.

(c) 6 violations in 77174 transitions (0.0077% violation rate )for our algorithm.

Figure 5: State visiting and constraint violation for $\epsilon_{risk} = 1.25$.

#### B.3.2    CONTINUOUS GRIDWORLD

For the problem setting, when the agent takes one step toward a direction, it will move along that direction with the distance following a uniform distribution $U(0.6, 1.2)$. As described in 5.2, for a transition $(s, a, s')$, the cost function penalizes proximity to the dangerous circles, defined as

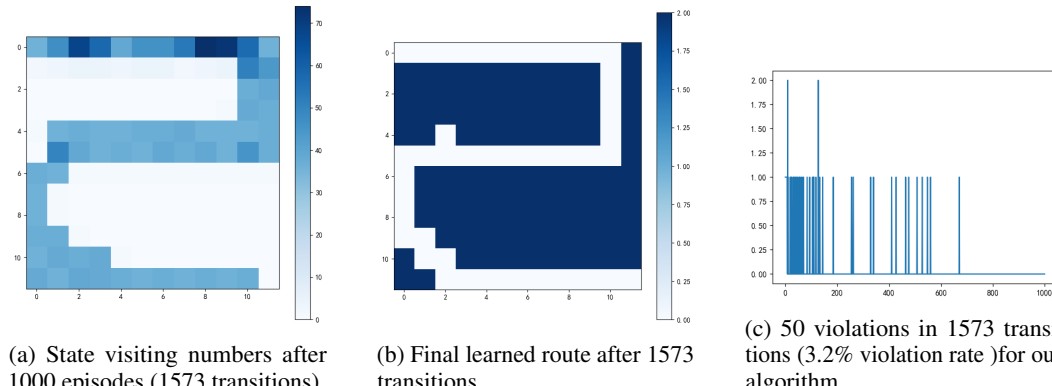

(a) State visiting numbers after 1000 episodes (1573 transitions).

(b) Final learned route after 1573 transitions.

(c) 50 violations in 1573 transitions (3.2% violation rate )for our algorithm.

Figure 6: State visiting and constraint violation for $\epsilon_{risk} = 0.25$.

$\max(4 - 2d_1, 4 - 2d_2, 0)$, where $d_1$ and $d_2$ denote Euclidean distances from the agent's next position $s'$ to the centers of two hazardous circles located at $(3, 3)$ and $(7, 7)$ with radius 2. Reward is defined as $\max\{10(1 - dist(s, (10, 10)), 0)\}$, which is positive exclusively in the upper-right corner and diminishes with distance from the goal. We use replay buffers to store the data collected and set the batch size to be 128. Other detailed parameters for continuous Gridworld are set as below: discounted factor $\gamma = 0.99$, the random probability for the policy starting from $\epsilon_p = 0.9$ and ending at 0.1, the stepsize for Q-value update $\beta_n$ is 0.005, the total time horizon $T = 180$, cost threshold $\epsilon_{risk} = 0.3$, the parameter for exploration set $\alpha_n$ is defined to be 0.7, the number of episodes is 250. As GP update will become slower when data are collected more, we discretize the state space into $20 * 20$ identical grids, and choose the grid centers as representer points for GP prediction. So now we make GP on 400 points. What's more, we use the penalized reward $r(s, a) - 50 \cdot c(s, a, s')$ for Q-learning training.

### B.3.3 CARTPOLE

The state of CartPole is a four-dimensional vector [Cart Position, Cart Velocity, Pole Angle, Pole Angular Velocity]. The cart position can take values between $(-4.8, 4.8)$, but the episode terminates if the cart leaves the $(-2.4, 2.4)$ range. The pole angle can be observed between $(-0.418, 0.418)$ but the episode terminates if the pole angle is not in the range $(-0.2095, 0.2095)$. The action space $\mathcal{A} = \{0, 1\}$, which means pushing cart to the left or right respectively. When the episode does not terminate, the reward is 1 for each step. Denote $p$ and $w$ to be the position and pole angle of the pole. Then the cost is defined to be $\max\{\text{dist}(p, [-1.9, 1.9]), 10 \cdot \text{dist}(w, [-0.15, 0.15])\}$, which is the maximum of distance (with scaling parameter) of position and pole angle away from a safe interval.

For parameters, we run 2000 episodes, and allow the maximum steps per episode to be 200. Random policy parameter $\epsilon_p$ starts at 0.2 and increases to 0.4 gradually. The discounted factor $\gamma = 0.99$ for Q-learning part, and the batch size for training is 100. Cost threshold $\epsilon_{risk} = 0.3$, and we use the penalized reward $r - 50 * c$ for Q-learning training. In our algorithm, we use two networks to predict the mean and variance of the cost simultaneously, which replaces GP for continuous state space. For ActSafe, there are two stages. In the first stage, the agent tries to reduce the uncertainty in GP for pair model, where we spend 40 episodes. In the second stage, the agent tries to maximize the expected cumulative reward, where we spend 160 episodes.

### B.3.4 SAFETY-GYMNASIUM

The task SafetyPointGoal1-v0 in the environment Safety-Gymnasium is shown in Figure7.

Episodes terminate upon success or time limit. Rewards encourage progress and success; an instantaneous safety cost $c_t \in \{0, 1\}$ is incurred when the robot intersects any hazard. Aligned with our almost-sure per-step safety objective, we adopt a per-step threshold $\varepsilon_{risk}$ decreasing from 0.8 to 0.2,. The reward is $r_t = (D_{\text{last}} - D_{\text{now}}) \beta$, where $D_{\text{last}}$ denotes the distance between the agent and Goal at the previous time step, $D_{\text{now}}$ denotes the distance between the agent and Goal at the current time step, and $\beta$ is a discount factor.

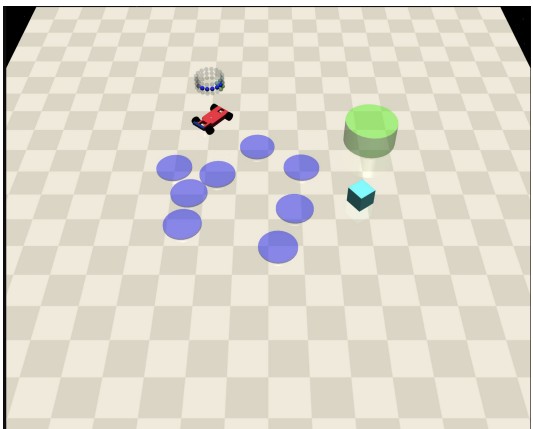

Figure 7: SafetyPointGoal1-v0 Task

We use the default time limit $T$=1000 per episode and discount $\gamma$=0.99. Training runs for 200 epoches with 4096 steps per episode. We seed the exploration set $\mathcal{S}_0$ by verifying small action-magnitude neighborhoods around initial states; only $(s, a)$ pairs with zero observed cost enter $\mathcal{S}_0$. The confidence multiplier $\alpha_n$ is annealed from optimistic to conservative milder values ( $\alpha_1$=0.2 $\rightarrow$ $\alpha_N$=1.1) to gradually expand $\mathcal{S}_n$. The UCB weight grows from 0.2 to 1.1. Also, we apply a penalized reward $r - \kappa c$ to train the Q-netwrok with an adaptive $\kappa$. To find the action that maximizes the Q-value among the continuous action space, we use cross-entropy method under the control of UCB safe set. For the cost prediction model loss, we use BCEWithLogitsLoss, where

$$\text{BCEWithLogits}(x, y) = -[y \log \sigma(x) + (1 - y) \log(1 - \sigma(x))]$$

for a given logits $x$ (unnormalized scores, any real number) and targets $y \in [0, 1]$.

All details can be found in safe_fqi.py file.

