# OpenReview forum: "Safe Learning Through Controlled Expansion of Exploration Set"
_ICLR.cc/2026/Conference — Submitted to ICLR 2026_

### Official Review · Reviewer_MZgE · 2025-10-20

**Soundness:** 1
**Presentation:** 2
**Contribution:** 1
**Rating:** 2
**Confidence:** 5

**Summary:**

This paper studies safe reinforcement learning (Safe RL) under unknown safety constraints and proposes an algorithm that guarantees almost-sure safety by gradually expanding an exploration set based on Gaussian Process (GP) predictions. While the topic is relevant and important, I find that the paper does not meet the scientific and originality standards expected at ICLR. The main concerns are the lack of novelty, insufficient discussion of related work, and limited empirical evaluation.

**Strengths:**

1. This paper is easy to follow. Motivations behind this paper is clearly explained.
1. The proposed method is technically sound.

**Weaknesses:**

1. Several existing studies have already addressed Safe RL with almost-sure safety guarantees, such as Turchetta et al. (2016) and Wachi et al. (2020). These works also model the safety cost function with Gaussian Processes and define safe regions based on pessimistic predictions. The present paper does not cite or discuss these prior studies, nor does it explain clearly what distinguishes its setting or contributions.

    - Turchetta, Matteo, Felix Berkenkamp, and Andreas Krause. "Safe exploration in finite markov decision processes with gaussian processes." Advances in neural information processing systems 29 (2016).
    - Wachi, Akifumi, and Yanan Sui. "Safe reinforcement learning in constrained markov decision processes." International Conference on Machine Learning. PMLR, 2020.

1. Although the authors may want to argue that the above previous works assume deterministic transitions or state-only safety functions, they do not analyze how these assumptions fundamentally change the problem or the level of difficulty. A more detailed comparison and discussion are necessary to clarify the true novelty of this work.

1. The core ideas (i.e., using a GP to predict the safety cost function and defining a safe region via upper confidence bounds) are already established in the above literature. The proposed method seems to be a minor variation rather than a conceptual advancement. The paper should better articulate what technical challenge is being solved here that was not addressed by earlier approaches.

1. The regret bound and convergence results appear to be straightforward extensions of existing results for safe exploration and constrained MDPs. The derivations largely follow standard arguments, and no genuinely new theoretical insight is provided. The theoretical section does not contribute sufficient novelty to justify publication.

1. The experimental evaluation is not convincing. Overall, the experiments are too limited to support the claims of effectiveness.
    - The comparison with ActSafe is limited to a simple Gridworld toy example.
    - The results on the Safety-Gymnasium benchmark include only CPO as a baseline. However, CPO handles different types of constraints, so the comparison is not entirely fair or sufficient.
    - While Safety-Gymnasium is a suitable benchmark, a broader set of baselines are needed to demonstrate the advantage of the proposed method.

**Questions:**

I do not have any question.

In summary, the paper lacks sufficient novelty in both algorithmic and theoretical aspects. The related work section does not adequately situate the contribution in the context of prior Safe RL research, and the experimental results are not strong enough to justify publication. I therefore do not recommend acceptance at ICLR.

---

> ### Author Response · Authors · 2025-12-02
>
> **Question**:
>
> Several existing studies have already addressed Safe RL with almost-sure safety guarantees, such as Turchetta et al. (2016) and Wachi et al. (2020). These works also model the safety cost function with Gaussian Processes and define safe regions based on pessimistic predictions. The present paper does not cite or discuss these prior studies, nor does it explain clearly what distinguishes its setting or contributions.
>
> [1] Turchetta, Matteo, Felix Berkenkamp, and Andreas Krause. "Safe exploration in finite markov decision processes with gaussian processes." Advances in neural information processing systems 29 (2016).
>
> [2] Wachi, Akifumi, and Yanan Sui. "Safe reinforcement learning in constrained markov decision processes." International Conference on Machine Learning. PMLR, 2020.
>
> **Response**：
>
> Thanks for your helpful comment. We will add the following discussion in our paper:
>
> [1], [2] rely on a strong assumption that the deterministic transition function is known, so they can directly find a path to any state they want to evaluate. In practice, they want to evaluate the state with maximum uncertainty under GP.
> However, our paper
> considers an unknown stochastic transition function and needs to explore under this unknown environment.
> Our method gives out an expected number of transitions to fully explore all state-action pairs through an $\epsilon-$ greedy exploration policy.
>
>
> **Question**:
>
> Although the authors may want to argue that the above previous works assume deterministic transitions or state-only safety functions, they do not analyze how these assumptions fundamentally change the problem or the level of difficulty. A more detailed comparison and discussion are necessary to clarify the true novelty of this work.
>
> **Response**:
>
> Thanks for your helpful comments.
> We will add a paragraph detailing  how unknown state-action costs change the problem or the level of difficulty.
>
>
> Switching from deterministic dynamics with state-only safety $c(s)$ to stochastic transitions with state-action safety $c(s, a)$  raises the difficulty of the problem class: safety must be certified before execution for each proposed ($s, a$) (not just for states), because randomness in $s_{t+1}$ can turn optimistic moves into unsafe trajectories;  safe-set growth must consider the full expansion depending on random trajectories; and performance guarantees must decompose regret, adding an explicit expansion cost (discovering the full safe region ) to the regret of standard sub-MDP learning.
>
>
> **Question**:
>
> The core ideas (i.e., using a GP to predict the safety cost function and defining a safe region via upper confidence bounds) are already established in the above literature. The proposed method seems to be a minor variation rather than a conceptual advancement. The paper should better articulate what technical challenge is being solved here that was not addressed by earlier approaches.
>
>
>
> **Response**:
>
> Thanks for your helpful comment.
> Previous work like don't show  how to explore unvisited state-action pairs to reduce the confidence interval
> of GP. It should be noticed that our paper considers this exploration process and shows an expected number of transitions to fully explore all state-action pairs through an $\epsilon$-greedy exploration policy,
> which is shown in the proof of Theorem 1. We will hightlight this theoretical novelty.
>
> **Question**:
>
> The regret bound and convergence results appear to be straightforward extensions of existing results for safe exploration and constrained MDPs. The derivations largely follow standard arguments, and no genuinely new theoretical insight is provided. The theoretical section does not contribute sufficient novelty to justify publication.
>
>
> **Response**:
>
> Thanks for your helpful comments. We’ll  highlight the theoretical novelty.
> Our method gives out an expected number of transitions to fully explore all state-action pairs through an $\epsilon-$ greedy exploration policy.
> Based on this result, the regret can be decomposed into two terms. The first term quantifies how much expansion delays optimal safe learning, and the second term is about the almost sure convergence of the learned policy conditioned on full exploration. This decomposition is our new result.

---

> > ### Author Response · Authors · 2025-12-02
> >
> > **Question**:
> >
> > The experimental evaluation is not convincing. Overall, the experiments are too limited to support the claims of effectiveness.
> >
> > - The comparison with ActSafe is limited to a simple Gridworld toy example.
> >
> > - The results on the Safety-Gymnasium benchmark include only CPO as a baseline. However, CPO handles different types of constraints, so the comparison is not entirely fair or sufficient.
> >
> > - While Safety-Gymnasium is a suitable benchmark, a broader set of baselines are needed to demonstrate the advantage of the proposed method.
> >
> >
> > **Response**:
> >
> > Thanks for your insightful comments.
> > Four experiment environments are designed from low level to high level. The state and action spaces are both discrete in discrete Gridworld, and we extend to continuous state space in continuous Gridworld. In Cartpole we consider both the continuous state and actions spaces, and in  PointGoal1 we consider the high-dimensional continuous task, which is most complicated. By organize environments in this order, we want to show the lower training
> > costs and higher final reward in our theoretical results and demonstrate the scalability.
> > We will add comparison against more baseline in safety-Gymnasium task.

---

### Official Review · Reviewer_QSBx · 2025-10-25

**Soundness:** 2
**Presentation:** 2
**Contribution:** 1
**Rating:** 2
**Confidence:** 4

**Summary:**

This paper proposes a safe exploration method with a predictive Gaussian process model. Asymptotic convergence to the optimal policy and a bound on online regret are provided.

**Strengths:**

The addressed problem is important, and the presentation of the paper is fair.

**Weaknesses:**

**Weaknesses:**
1. The framework of safe exploration and expansion with a predictive Gaussian Process (GP) model is not novel; similar ideas have already been explored in [1]. Moreover, previous works [1–3] have addressed the case without any prior knowledge of the safety or cost functions. None of these studies is cited or discussed in the related works, which gives the impression that the authors are not fully aware of existing research in this area.
2. The theoretical results presented in Theorem 1 and Theorem 2 appear to be very similar to those in [1]. However, the paper does not clearly articulate how these results differ from or improve upon prior work. This omission raises concerns about the originality and depth of the theoretical contribution.
3. The proposed method does not effectively address safety during exploration, a key issue that has already been handled in prior GP-based frameworks such as [2]. As a result, the claimed novelty in “safe exploration” seems unsubstantiated.
4. The experimental evaluation is limited and lacks diversity. The gridworld environment is overly simplistic, while CartPole and PointGoal1 are of comparable complexity and fail to demonstrate scalability. The experimental section also lacks logical organization and a comprehensive summary. Since all case studies show similar trends, presenting them separately in the main text seems unnecessary.

[1] A. Wachi, Y. Sui, Safe reinforcement learning in constrained Markov decision processes. ICML 2020.
[2] A. Wachi, and et al, Safe exploration in reinforcement learning: a generalized formulation and algorithms, NeurIPS 2023.
[3] Akifumi Wachi, Wataru Hashimoto, Kazumune Hashimoto, Long-term Safe Reinforcement Learning with Binary Feedback, AAAI Conference on Artificial Intelligence (AAAI), 2024.

**Questions:**

Given the limited theoretical contribution and the lack of comprehensive experimental validation, I do not have further questions for the authors.

---

> ### Author Response · Authors · 2025-12-02
>
> **Question**:
>
> The framework of safe exploration and expansion with a predictive Gaussian Process (GP) model is not novel; similar ideas have already been explored in [1]. Moreover, previous works [1-3] have addressed the case without any prior knowledge of the safety or cost functions. None of these studies is cited or discussed in the related works, which gives the impression that the authors are not fully aware of existing research in this area.
>
> **Response**:
>
> Thanks for your helpful comment. We will integrate an explicit related work paragraph on GP-based safe expansion.  [1] relies on a strong assumption that the deterministic transition function is known, so they can directly find a path to any state they want to evaluate. [2], [3] and our paper all consider an unknown stochastic transition function. It should be noticed that [2] only states that if the GP prediction is accurate enough, then they can find the optimal policy. Their method lacks the exploration guarantee to make GP converge to the desired accuracy, which is important in cost prediction.  However, our method gives out an expected number of transitions to fully explore all state-action pairs through an $\epsilon-$greedy exploration policy.
> What's more, [3] relies on a key assumption that the cost function admits a linear approximation and estimate the linear parameter by a Maximum Likelihood estimator. We consider a GP method and don't need the specific linear structure.
>
>
>
> **Question**:
>
> The theoretical results presented in Theorem 1 and Theorem 2 appear to be very similar to those in [1]. However, the paper does not clearly articulate how these results differ from or improve upon prior work. This omission raises concerns about the originality and depth of the theoretical contribution.
>
>
> **Response**:
>
> Thanks for your helpful comment. The key difficulty  in safety control during training is how to control the safety as well as make effective exploration. [2] can achieve high-probability safe training because they use the confidence interval of GP. Their convergence result of the optimal policy is based on the assumption that the confidence interval of GP has been small enough. However, they don't show how to explore unvisited state-action pairs to reduce the confidence interval of GP. It should be noticed that our paper considers this exploration process and  shows an expected number of transitions to fully explore all state-action pairs through an $\epsilon-$ greedy exploration policy, which is shown in the proof of Theorem 1. We will highlight the theoretical novelty.
>
>
>
> **Question**:
>
> The proposed method does not effectively address safety during exploration, a key issue that has already been handled in prior GP-based frameworks such as [2]. As a result, the claimed novelty in "safe exploration" seems unsubstantiated.
>
> **Response**:
>
> Thanks for your insightful comments.
> By  section 3.4, each trajectory in policy deployment terminates if the current state-action pair is outside $S_n$, or the length of trajectory is larger than $T$, or safety constraint is violated more than once. So each trajectory  only contains at most one violation. Notice that if we choose $\alpha_n$ to be some constants related to  the maximal mutual information about GP posterior and any $\delta\in(0,1)$, $(\mu_n-\alpha_n \sigma_n,\mu_n+\alpha_n \sigma_n)$ will be a $1-\delta$ confidence interval of the true cost function for all pairs by convergence result of GP in [5]. If we delete the terminal condition of one violation,  then the probability each trajectory contain at most $k$ violations is  at least $\sum_{i=0}^k\binom{T}{i} (1-\delta)(1-\epsilon_p)^{T-i}\epsilon_p^i$. Especially,  the probability each trajectory contain zero violations is  at least $ (1-\delta)(1-\epsilon_p)^{T}$.
> Since each unsafe pair will be visited only once and never be chosen again by our policy, the total violation number is the total number of unsafe pairs.
>
> It should be noticed that there is a tradeoff between safety violation and policy optimization. If we want to learn the optimal safe policy, we have to learn the environment completely and train the policy inside the safe region that has been learnt.
>
> [5] Chowdhury S R, Gopalan A. On kernelized multi-armed bandits[C]//International Conference on Machine Learning. PMLR, 2017: 844-853.

---

> > ### Author Response · Authors · 2025-12-02
> >
> > **Question**:
> >
> > The experimental evaluation is limited and lacks diversity. The gridworld environment is overly simplistic, while CartPole and PointGoal1 are of comparable complexity and fail to demonstrate scalability. The experimental section also lacks logical organization and a comprehensive summary. Since all case studies show similar trends, presenting them separately in the main text seems unnecessary.
> >
> >
> >
> >
> >
> > [1] A. Wachi, Y. Sui, Safe reinforcement learning in constrained Markov decision processes. ICML 2020. [2] A. Wachi, and et al, Safe exploration in reinforcement learning: a generalized formulation and algorithms, NeurIPS 2023. [3] Akifumi Wachi, Wataru Hashimoto, Kazumune Hashimoto, Long-term Safe Reinforcement Learning with Binary Feedback, AAAI Conference on Artificial Intelligence (AAAI), 2024.
> >
> > **Response**:
> >
> > Thanks for your insightful comments.
> > Four experiment environments are designed from low level to high level. The state and action spaces are both discrete in discrete Gridworld, and we extend to continuous state space in continuous Gridworld. In Cartpole we consider both the continuous state and actions spaces, and in  PointGoal1 we consider the high-dimensional continuous task, which is most complicated. By organize environments in this order, we want to show the lower training
> > costs and higher final reward in our theoretical results and demonstrate the scalability.

---

### Official Review · Reviewer_UKkz · 2025-11-01

**Soundness:** 2
**Presentation:** 2
**Contribution:** 2
**Rating:** 2
**Confidence:** 3

**Summary:**

This paper introduces LearnSEES, a novel algorithm for Safe Reinforcement Learning in an online episodic setting. The core objective is to maximize cumulative reward subject to an almost-sure safety constraint, meaning every single transition must satisfy $c(s_t, a_t) \le \epsilon_{\text{risk}}$. Crucially, the cost function $c(s, a)$ is assumed to be unknown. The method addresses this by starting with an initial safe set and iteratively expanding it. At each episode, a Gaussian Process (GP) is trained on collected cost data to provide a pessimistic upper confidence bound (UCB) on the cost for unvisited state-action pairs. This bound is used to define a new, larger safe exploration set $S_n$. Within this set, a policy is learned using a UCB-augmented Q-learning approach, which explicitly penalizes observed unsafe transitions. The authors provide theoretical guarantees for asymptotic convergence to the optimal safe policy and a finite-episode online regret bound. Experiments on Gridworld, CartPole, and Safety-Gymnasium benchmarks demonstrate superior safety performance during training compared to baselines.

**Strengths:**

The paper presents a novel combination of techniques to enforce a very strong safety guarantee. The use of a GP-based pessimistic UCB to define a progressively expanding, almost-surely safe exploration set $S_n$ is a creative and sound approach to tackle the unknown safety constraint problem. While the concept of safe set expansion is not new (e.g., Berkenkamp et al., 2017), the integration with finite-horizon Q-learning and the explicit focus on almost-sure safety for every transition during the entire learning phase is a valuable contribution to the Safe RL literature.

The paper is well-structured, and the technical development is rigorous. The theoretical analysis, including the asymptotic convergence proof (Theorem 4.1) and the online regret bound (Theorem 4.2), provides a solid foundation for the proposed algorithm. The explicit handling of unsafe transitions by setting $Q(s, a) = -1$ is a clean, practical mechanism within the Q-learning framework to discourage high-risk exploration.

**Weaknesses:**

1. The entire framework hinges on the use of a Gaussian Process (GP) to model the cost function $c(s, a)$ and provide reliable uncertainty estimates. While GPs are excellent for low-dimensional, continuous problems, they are notoriously non-scalable. The computational complexity of GP inference is typically $O(N^3)$, where $N$ is the number of data points. For any realistic robotics task (e.g., manipulation, locomotion) with high-dimensional state and action spaces, the number of required samples $N$ quickly becomes intractable. The authors mention that other models could be substituted if they provide epistemic uncertainty, but the theoretical guarantees rely directly on the GP's properties (e.g., the confidence bound in Eq. 3). This reliance severely limits the practical applicability of LearnSEES to the very domain (robotics) that motivates the strong safety guarantees.

2. The theoretical analysis and the Q-learning update (Eq. 4) implicitly assume a finite or discretizable state-action space, which is a standard limitation for UCB-style Q-learning. While the GP handles continuous spaces, the overall framework, especially the regret analysis, is more aligned with tabular or heavily discretized settings. The paper needs to be more explicit about how the algorithm handles continuous state-action spaces in the Q-learning and exploration phases, beyond just the GP prediction.

3. The experiments, while demonstrating the safety advantage, are conducted on relatively simple environments (Gridworld, CartPole, and a few Safety-Gymnasium tasks). To truly validate the method's significance for robotics, a demonstration on a more complex, high-dimensional continuous control task (e.g., a challenging MuJoCo environment or a real-world system) is essential. The current results, while positive, do not fully alleviate the concerns about scalability and the GP's performance in complex domains.

**Questions:**

1. The $O(N^3)$ complexity of the GP is a major bottleneck. Can the authors elaborate on how they envision scaling this approach to high-dimensional continuous control problems (e.g., 10+ state dimensions, 3+ action dimensions)? Have the authors considered sparse GP approximations or deep kernel learning to maintain the uncertainty estimates while improving scalability? If so, how would the theoretical guarantees (Theorems 4.1 and 4.2) be affected by the approximate nature of these models?

2. The policy update uses $\max_{a' \in A} Q_t^n(s', a')$ in Equation 4, which is characteristic of discrete action spaces. Given that CartPole and Safety-Gymnasium are often treated as continuous control problems, how is the continuous action space handled in practice? Is the action space discretized, and if so, how does the discretization density affect the almost-sure safety guarantee?

3. The algorithm requires an initial safe exploration set $S_0$. In real-world applications, identifying a non-trivial $S_0$ that is guaranteed to be safe can be a significant challenge, often requiring expert knowledge or extensive pre-testing. Can the authors discuss the sensitivity of LearnSEES to the size and quality of $S_0$? For instance, what happens if $S_0$ is too small or, worse, contains a few unverified unsafe points?

4. While the paper correctly notes that CMDP methods enforce constraints in expectation, a more direct empirical comparison to state-of-the-art model-free Safe RL algorithms (e.g., CPO, PPO-Lagrangian) on the Safety-Gymnasium tasks would strengthen the paper. Specifically, a plot showing the instantaneous constraint violation (not just cumulative) for LearnSEES vs. a well-tuned CMDP method would clearly highlight the benefit of the almost-sure safety approach during the early, critical phases of learning.

---

> ### Author Response · Authors · 2025-12-02
>
> **Question**:
>
> The $O(N^3)$ complexity of the GP is a major bottleneck. Can the authors elaborate on how they envision scaling this approach to high-dimensional continuous control problems (e.g., $10+$ state dimensions, $3+$ action dimensions)? Have the authors considered sparse GP approximations or deep kernel learning to maintain the uncertainty estimates while improving scalability? If so, how would the theoretical guarantees (Theorems 4.1 and 4.2 ) be affected by the approximate nature of these models?
>
> **Response**:
>
> Thanks for your insightful comments and questions. For high-dimensional continuous control problems, we use neural network to predict the mean and variance of cost function. Theorems 4.1 and 4.2 only applies for tabular setting since we consider the full exploration of all state-action pairs.
>
>
>
>
> **Question**:
>
> The policy update uses $\max _{a^{\prime} \in A} Q_t^n\left(s^{\prime}, a^{\prime}\right)$ in Equation 4, which is characteristic of discrete action spaces. Given that CartPole and Safety-Gymnasium are often treated as continuous control problems, how is the continuous action space handled in practice? Is the action space discretized, and if so, how does the discretization density affect the almost-sure safety guarantee?
>
> **Response**:
>
> Thanks for your insightful comments and questions.
> For continuous tasks we (i) keep the safety gate continuous by acting only on actions satisfying $\mu_n(s, a)+\alpha_n \sigma_n(s, a) \leq \varepsilon_{\text {risk }}$, and (ii) compute the Q-update's greedy action via a continuous optimization called cross-entropy method(CEM) under that gate. In SafetyGymnasium we implemented this with a neural cost model that outputs ($\mu, \sigma^2$) and used CEM to approximate $\arg \max _a Q\left(s^{\prime}, a^{\prime}\right)$ while enforcing the UCB constraint. We will clarify this in Section 5.4 and Appendix.
>
>
>
> **Question**:
>
> The algorithm requires an initial safe exploration set $S_0$. In real-world applications, identifying a non-trivial $S_0$ that is guaranteed to be safe can be a significant challenge, often requiring expert knowledge or extensive pre-testing. Can the authors discuss the sensitivity of LearnSEES to the size and quality of $S_0$ ? For instance, what happens if $S_0$ is too small or, worse, contains a few unverified unsafe points?
>
>
>
> **Response**:
>
> Thanks for your insightful questions.
> If $S_0$ is tiny, expansion proceeds (more slowly) via the UCB frontier; if $S_0$ contains an unsafe pair, our deployment rule marks it "verified-unsafe" and prunes it from $S_n$.
>
>
>
>
>
> **Question**:
>
> While the paper correctly notes that CMDP methods enforce constraints in expectation, a more direct empirical comparison to state-of-the-art model-free Safe RL algorithms (e.g., CPO, PPOLagrangian) on the Safety-Gymnasium tasks would strengthen the paper. Specifically, a plot showing the instantaneous constraint violation (not just cumulative) for LearnSEES vs. a welltuned CMDP method would clearly highlight the benefit of the almost-sure safety approach during the early, critical phases of learning.
>
> **Response**:
>
> Thanks for your insightful questions.
> We already compare to CPO and report per-episode cost histograms.  In the future we will add (i) instantaneous violation rates over time (per step), and (ii) PPO-Lagrangian tuned on the same seeds, both on SafetyPointGoal1.

---

### Official Review · Reviewer_gsgY · 2025-11-04

**Soundness:** 2
**Presentation:** 3
**Contribution:** 3
**Rating:** 4
**Confidence:** 4

**Summary:**

This paper proposes a model-free Safe Reinforcement Learning (Safe RL) algorithm, LearnSEES, that aims to ensure safe exploration by progressively expanding an exploration set of verified safe state–action pairs. A Gaussian Process (GP) model predicts the cost and uncertainty of unexplored pairs, and the safe region is expanded cautiously based on upper confidence bounds. Learning occurs only within this safe set via Q-learning with an exploration bonus.
The authors prove asymptotic convergence to the optimal policy within the safe sub-MDP (Theorem 1) and establish a finite-time regret bound of O(logN) (Theorem 2). Experiments on several benchmarks demonstrate that the method attains low violation rates and near-optimal performance compared with baselines.

**Strengths:**

+ Introduces an incremental safe-set expansion mechanism using GP uncertainty estimates.
+ Provides asymptotic and finite-time theoretical analyses that link safe exploration to standard Q-learning bounds.
+ Achieves low empirical violation rates in both discrete and continuous benchmarks.

**Weaknesses:**

-  Violations may still occur during training and are merely controlled per episode.   The theory only supports a high-probability bound on the number of violations created by the Gaussian Process (GP), not a zero-violation or truly almost-sure guarantee.

- Equation (3) defines the exploration set as
$
    S_n = \{(s,a) : \max_a \mu_n(s,a) + \alpha_n k_n(s,a;s,a) \le \epsilon_{\text{risk}}\},
$
    which requires all actions at a state to be safe. This contradicts Definition~3.1, which states that a state is safe if there exists at least one safe action.

- The paper does not define the noise model, kernel family, or confidence parameter $\alpha_n$ that ensure a uniformly valid upper confidence bound (UCB) over all  queried $(s,a)$ pairs.  The proofs of Theorems~1--2 do not condition on or invoke the GP concentration events, meaning the claimed probabilistic safety guarantees are not rigorously established.

- The regret proof builds on Yang et~al. (2021), which assumes specific step-size and UCB schedules not verified for the proposed implementation.

**Questions:**

How is “almost-sure safety” defined formally?  Is it per step, per episode, or over all training trajectories?

How are the GP kernel, noise variance, and $\alpha_n$ selected to achieve valid uncertainty bounds?

How does the regret bound scale if the GP misclassifies a safe pair as unsafe?

---

> ### Author Response · Authors · 2025-12-02
>
> **Question**:
>
>  How is "almost-sure safety" defined formally? Is it per step, per episode, or over all training trajectories?
>
> **Response**:
>
> Thanks for your insightful questions.
> Our "almost-sure safety" is defined for per-step constraint: $c\left(s_t, a_t\right) \leq \varepsilon_{\text {risk }}$ for all $t$ (Problem Formulation, Eq. (1)). Our theorem about "almost sure" concerns policy convergence on the safe sub-MDP, not zero violations. Exploration safety is enforced with high probability via the UCB gate that builds the exploration set. We'll revise wording to avoid conflating these: "a.s." refers to convergence of $\pi_n$, while safety during training is controlled with high probability by the confidence schedule.
>
> **Question**:
>
> How are the GP kernel, noise variance, and $\alpha_n$ selected to achieve valid uncertainty bounds?
>
> **Response**:
>
> Thanks for your insightful question.
> We will state the following in the paper: (i) kernel $k$ : squared-exponential with bounded RKHS norm on $c(\cdot)$; (ii) No observation noise since we consider deterministic costs; (iii) $\alpha_n$ will gradually decreases when the uncertainty reduces. These assumptions will be added to paper directly.
>
>
> **Question**:
>
>  How does the regret bound scale if the GP misclassifies a safe pair as unsafe?
>
>
> **Response**:
>
> Thanks for your insightful question.
> A temporary false negative delays discovery of some safe $(s, a)$. In our analysis this is reflected in the prediscovery constant term (the "cost of learning the safe structure")  $T \cdot \frac{|S|^2|A|^2}{\epsilon_p \delta}$ in Theorem 2; once the set is recovered, the regret remains $O(\log N)$ from tabular Q-learning on the safe sub-MDP.

---

### Meta-Review · Area_Chair_rmK5 · 2026-01-04

**Summary:**

**Summary of the paper**

This paper proposes LearnSEES, a model-free safe reinforcement learning algorithm for settings where state–action safety constraints are unknown a priori. The method maintains an exploration set of verified safe state–action pairs and gradually expands this set using pessimistic upper confidence bounds derived from a Gaussian Process over the cost function. Policy learning is restricted to the current safe set via a UCB-style Q-learning procedure, and the authors provide asymptotic convergence guarantees to the optimal policy in the induced safe sub-MDP along with a finite-episode regret bound. Experimental results on Gridworld, CartPole, and Safety-Gymnasium demonstrate reduced safety violations during training compared to selected baselines. The paper positions its main contribution as achieving stronger (per-step) safety guarantees during learning under unknown costs.

**Summary of the reviewers' concern**

The primary concerns raised by multiple reviewers center on lack of novelty, insufficient engagement with prior work, and limited empirical validation, with these issues articulated most clearly and convincingly in Review 4. Several reviewers note that the core ideas, GP-based modeling of unknown safety costs, pessimistic confidence bounds, and safe-set expansion, are well established in prior Safe RL literature (e.g., Turchetta et al., 2016; Wachi & Sui, 2020), yet the paper does not adequately situate itself relative to these works. Theoretical guarantees (asymptotic convergence and regret bounds) are viewed as largely straightforward extensions of existing analyses, without introducing genuinely new conceptual or technical insights. Reviewers also express concern that the experimental evaluation is too limited and does not convincingly demonstrate scalability or practical relevance beyond relatively simple benchmarks, especially given the reliance on GP-style uncertainty modeling.

**Reviewer Concerns:**

The rebuttal only partially addresses concerns regarding missing related work and clarifies how the authors view their setting as distinct from prior GP-based safe exploration frameworks, particularly by emphasizing unknown stochastic transitions and explicit exploration to shrink GP uncertainty. However, key concerns remain outstanding. Most importantly, the rebuttal does not convincingly resolve the novelty gap identified by Reviews 3 and 4. While distinctions relative to prior work are asserted, they are incremental in nature and not shown to fundamentally change the problem class or introduce new algorithmic principles. The theoretical contributions remain close to existing results, and the rebuttal does not provide a compelling argument for why these constitute a substantive advance. Additionally, concerns about experimental depth and scalability, particularly the reliance on relatively simple environments and limited baselines, are only acknowledged and deferred to future work.

**Reviewer Scores:**

The rebuttal does not address the concerns of the reviewers (please see above). So, the reviewers are NOT likely to change the score.

---

### Decision · Program_Chairs · 2026-01-26

Reject